# Maintenance over Time of the Effect Produced by Esmolol on the Structure and Function of Coronary Arteries in Hypertensive Heart Diseases

**DOI:** 10.3390/antiox11102042

**Published:** 2022-10-17

**Authors:** Raquel Martín-Oropesa, Pilar Rodríguez-Rodríguez, Laia Pazó-Sayós, Ana Arnalich-Montiel, Silvia Magdalena Arribas, Maria Carmen González, Begoña Quintana-Villamandos

**Affiliations:** 1Department of Anesthesiology, Hospital General Universitario Gregorio Marañón, 28007 Madrid, Spain; 2Department of Physiology, Faculty of Medicine, Universidad Autónoma de Madrid, 28029 Madrid, Spain; 3Department of Pharmacology and Toxicology Faculty of Medicine, Universidad Complutense de Madrid, 28040 Madrid, Spain

**Keywords:** esmolol, vascular remodeling, spontaneously hypertensive rat, treatment withdrawal

## Abstract

We previously observed that esmolol treatment for 48 h reduced vascular lesions in spontaneously hypertensive rats (SHRs). Therefore, we investigated whether this beneficial effect is persistent after withdrawal. Fourteen-month-old SHRs (SHR-Es) were treated with esmolol (300 μg/kg/min) or a vehicle for 48 h. Two separate groups were also given identical treatment, but they were then monitored for a further 1 week and 1 month after drug withdrawal. We analyzed the geometry and composition of the coronary artery, vascular reactivity and plasma redox status. Esmolol significantly decreased wall thickness (medial layer thickness and cell count), external diameter and cross-sectional area of the artery, and this effect persisted 1 month after drug withdrawal. Esmolol significantly improved endothelium-dependent relaxation by ACh (10^−9^–10^−4^ mol/L); this effect persisted 1 week (10^−9^–10^−4^ mol/L) and 1 month (10^−6^–10^−4^ mol/L) after withdrawal. Esmolol reduced the contraction induced by 5-HT (3 × 10^−8^–3 × 10^−5^ mol/L), and this effect persisted 1 week after withdrawal (10^−6^–3 × 10^−5^ mol/L). Esmolol increased nitrates and reduced glutathione, and it decreased malondialdehyde and carbonyls; this enhancement was maintained 1 month after withdrawal. This study shows that the effect of esmolol on coronary remodeling is persistent after treatment withdrawal in SHRs, and the improvement in plasma oxidative status can be implicated in this effect.

## 1. Introduction

Arterial hypertension is the most prevalent risk factor for cardiovascular disease. Every year, it is associated with high mortality and morbidity [1]. It has been widely demonstrated that hypertension induces structural and physiological changes in the arterial wall [2], and regression of these changes is associated with reduced incidence of cardiovascular events [3,4]. Therefore, we need drugs that interfere with the structural remodeling process and thus reducing endothelial dysfunction. Several studies have shown that chronic treatment with antihypertensive drugs produces regression of coronary artery remodeling [5,6,7,8,9]. However, our group has demonstrated this beneficial effect with short-term treatment with antiarrhythmic drugs (esmolol and dronedarone) [10,11,12,13].

Esmolol is an ultrashort-acting cardioselective β-adrenergic receptor antagonist with a different pharmacokinetic profile from that of classic β-blockers [14]. Esmolol has a cardioprotective effect [15]. In clinical practice, esmolol decreases myocardial ischemia [16], and, in preclinical studies, it increases nitric oxide bioavailability and reduces asymmetric dimethylarginine (ADMA) concentration in plasma (both related to vascular remodeling) [10,11].

We previously observed that esmolol treatment for 48 h produced reductions in blood pressure and heart rate in spontaneously hypertensive rats (SHRs) [10,11]. In addition, these effects were accompanied by an important regression of structural and functional remodeling of the coronary artery. Thus, the first aim of this study was to investigate whether the beneficial effect of esmolol on coronary artery remodeling persists after treatment withdrawal. Furthermore, this study aims to explore the implication of oxidative stress (plasma oxidative status) in the regression of vascular remodeling after esmolol withdrawal.

## 2. Material and Methods

### 2.1. Animal Model and Treatment

Fifty-four 14-month-old male SHRs were used in this study, and they were housed in cages (four per cage) under a 12 h/12 h light/dark cycle and maintained at a constant temperature of 24 °C and relative humidity of 40%. At the end of the treatment, the rats were sedated with 4 mg/kg diazepam and 10 mg/kg ketamine intraperitoneal injection, and blood was obtained from the tail vein in tubes with heparin (5000 units) to assess oxidative stress. Thereafter, the rats were killed by decapitation, and the heart was obtained. The left anterior descending coronary artery was then dissected to assess vascular function and structure.

SHRs were treated with an IV infusion of esmolol (Baxter, S.L.) at 300 μg/kg/min (SHR-Es) for 48 h. Another two separate groups were given identical treatment but were then monitored for an additional 1 week (SHR-E 7 d group) and 1 month (SHR-E 1 m group) after drug treatment withdrawal. SHRs receiving a vehicle (saline solution) were used as controls (SHR 48 h, SHR 7 d and SHR 1 m groups).

### 2.2. Blood Pressure and Heart Rate

Systolic blood pressure (SBP) and heart rate (HR) were measured using the tail-cuff method with a photoelectric sensor (Niprem 546, Cibertec, Madrid, Spain).

### 2.3. Wire Myography for the Study of Coronary Artery Vascular Reactivity

The heart was removed and maintained in cold (4 °C) oxygenated Krebs–Henseleit solution (KHS, in mmol/L: 115 NaCl, 25 NaHCO_3_, 4.7 KCl, 1.2 MgSO_4_⋅7H_2_O, 2.5 CaCl_2_, 1.2 KH_2_PO_4_, 11.1 glucose and 0.01 Na_2_EDTA). Then, 2 mm length segments of the left anterior descending coronary artery were mounted on a wire myograph (Multi Myograph System, model 610M, Danish Myo-Technology, Hinnerup, Denmark) as previously described [10,11], and data were recorded through a Powerlab data acquisition system (AD-Instruments, Castle Hill, Sydney, NSW, Australia). To assess endothelium-dependent relaxations, the segments were precontracted with 5-hydroxytryptamine (Sigma-Aldrich, St. Louis, MO, USA) (5-HT, 3 × 10^−7^ mol/L), and dose–response curves for acetylcholine (ACh 10^−9^–10^−4^ mol/L) were obtained. Relaxation was expressed as the percent of contraction reduction. After a 30 min washout period, contractile responses to serotonin were evaluated (10^−9^–3 × 10^−5^ mol/L). Contraction was expressed as a percentage of the maximum response of the arteries to K^+^-KHS.

### 2.4. Confocal Microscopy for the Study of Coronary Artery Structure and Composition

From the left anterior descending coronary artery, 1 mm length segments were isolated. From them, one ring and two longitudinal sections were cut and stained with nuclear dye DAPI (1:500 of a 5 mg/mL stock solution) and mounted on a slide with citifluor. One longitudinal section was mounted with the endothelial side up, and the other was mounted with the adventitial side up and visualized with a Leica TCS SP2 confocal system (Leica Microsystems, Wetzlar, Germany) to study the geometry and composition (cell number in the adventitia and media layers) as previously described [11]. To visualize cell nuclei, the 405 Ex/410–475 emission wavelengths were used. Three randomly selected regions were visualized with a ×20 objective at zoom 8 from each segment, and stacks of serial 1 μm optical sections were captured from the adventitial layer (longitudinal sections mounted with the adventitial side up) and medial layer (longitudinal sections mounted with the endothelial side up). The arterial rings were used to analyze arterial geometry. The rings were visualized with a ×20 objective with the 488 Ex/515 Em nm line showing internal and external diameters based on the autofluorescence of elastin, and from each ring, an image was captured.

Quantitative analysis of confocal data. MetaMorph Image Analysis Software (Universal Imaging, Wokingham, UK) was used as previously described [17]. Adventitial layer thickness was determined by the number of planes between the first and last images showing an adventitial cell nucleus at the maximum intensity. Similarly, medial layer thickness was assessed by the number of planes between the first and last image showing smooth muscle cells (SMC), since both adventitia and SMC and endothelial cells have distinctive shapes and orientations. Adventitial cells and SMCs were counted in a specific volume defined by the image area and the layer thickness of each particular vessel.

### 2.5. Biomarkers of Oxidative Status in Plasma

Blood samples (1.2 mL) with heparin were centrifuged at 900× *g* for 10 min at 4 °C, and the plasma was aliquoted and stored at −70 °C until use.

#### 2.5.1. Reduced Glutathione (GSH) Content

Plasma GSH was assessed using a fluorometric micromethod based on the reaction with o-phthalaldehyde (Sigma-Aldrich). Fluorescence was measured in a Synergy HT multimode microplate reader (Sinergy HT, BioTek, Winooski, VT, USA) at 360 ± 40 nm excitation and 460 ± 40 nm emission wavelengths, and GSH was quantified as µmol/mg protein [18].

#### 2.5.2. Nitrates

Plasma nitrates were assessed by Griess reaction, modified to a microplate reader. In brief [19], a 100 µL plasma sample was incubated with 10 µL N-ethylmaleimide 150 mM for 5 min at room temperature (Thermo Fisher Scientific, Waltham, MA, USA), and 110 µL trichloroacetic acid 20% *w*/*v* was added and centrifuged (12,000× *g*) for 5 min at 4 °C. A 40 µL volume of supernatant was transferred to a microplate, and the following reagents were added: 40 µL of vanadium (III) chloride (Sigma-Aldrich), 20 µL of sulfanilamide 2% (Thermo Fisher Scientific, Waltham, MA, USA) and 20 µL of naphthyl-ethylenediamine dihydrochloride 0.1% (Thermo Fisher Scientific, Waltham, MA, USA). The mixture was incubated for 1.0 h at 37 °C, and the absorbance was read at 540 nm. Nitrates were expressed as µM. The standard curve (0–100 µM) was prepared with a nitrate solution of 100 µM and diluted in methanol–H_2_O solution.

#### 2.5.3. Total Protein Carbonyls

Plasma protein carbonyls were assessed with a 2,4-dinitrophenylhydrazine-based assay, as previously described [18,20], and expressed as nanomoles per milligram (nmol/mg) of protein.

#### 2.5.4. Lipid Peroxidation

Plasma lipid peroxidation was evaluated by assessing the levels of the stable products malondialdehyde (MDA) and 4-Hydroxynonenal (HNE) using a commercial kit (Lipid Peroxidation Assay kit KB-03-002, Bioquochem, Gijon, Spain) as previously described [21]. MDA + HNE content was expressed as µM.

### 2.6. Statistical Analysis

Results are expressed as mean ± SEM. The parameters were compared using repeated-measures Student’s *t* test (for concentration–response curve parameters) and Student’s *t* test for independent samples (for physiological, structural and biochemical parameters). ACh and 5-HT vasoactive responses were expressed as differences in the areas under the concentration–response curves (AUCs) among the experimental groups. A Statistical significance level was established at *p*-values < 0.05.

## 3. Results

### 3.1. Blood Pressure and Heart Rate

A pronounced decrease in SBP and HR was observed after treatment with esmolol; however, these parameters returned to basal levels after withdrawal (Table 1).

### 3.2. Esmolol Improves Vascular Function, and This Effect Remains after Withdrawal

Esmolol significantly improved the endothelium-dependent relaxation induced by ACh (10^−9^–10^−4^ mol/L) in 5-HT contracted coronary arteries (Figure 1A). This improvement was maintained 1 week after withdrawal (ACh 10^−9^–10^−4^ mol/L) (Figure 1B) and 1 month after withdrawal (ACh 10^−6^–10^−4^ mol/L) (Figure 1C). The AUC was significantly higher in SHR-Es treated with esmolol than in SHRs (AUC_SHR-E 48 h_ = 209.6 ± 25.9 vs. AUC_SHR 48 h_ = 74.15 ± 8.67, *p* = 0.0001). This result was maintained 1 week after withdrawal (AUC_SHR-E 7 d_ = 185.16 ± 27.65 vs. AUC_SHR 7 d_ = 102.07 ± 18.45, *p* = 0.017), and 1 month after withdrawal (AUC_SHR-E 1 m_ = 136.33 ± 20.84 vs. AUC_SHR 1 m_ = 83.29 ± 11.87, *p* = 0.05).

Esmolol significantly reduced contraction induced by 5-HT (3 × 10^−8^–3 × 10^−5^ mol/L) in coronary arteries (Figure 1D). This improvement was maintained 1 week after withdrawal (10^−6^–3 × 10^−5^ mol/L) (Figure 1E). The AUC was significantly higher in SHRs than in SHR-Es treated with esmolol (AUC_SHR 48 h_ = 243.3 ± 19.66 vs. AUC_SHR-E 48 h_ = 28.1 ± 2.4, *p* = 0.00003), and this result was maintained 1 week after withdrawal (AUC_SHR 7 d_ = 133.13 ± 16.07 vs. AUC_SHR-E 7 d_ = 74.33 ± 8.68, *p* = 0.005). However, no difference was observed between the SHR 1 m and SHR-E 1 m groups (Figure 1F).

### 3.3. Esmolol Improves Vascular Structure, and This Effect Remains after Withdrawal

The effects of esmolol on coronary artery geometry are shown in Figure 2. External diameter (ED, internal diameter + adventitial layer + medial layer) (Figure 2B), wall thickness (WW, adventitial layer + medial layer) (Figure 2C) and cross-sectional area (CSA, adventitial layer + medial layer) (Figure 2D) of the coronary artery were significantly reduced after 48 h of treatment with esmolol (SHR-E 48 h group) with respect to untreated animals (SHR 48 h group). The beneficial effect of esmolol 300 µg/kg/min persisted after treatment withdrawal (SHR-E 7 d group vs. SHR 7 d group, and SHR-E 1 m group vs. SHR 1 m group) (Figure 2B–D). No differences in internal diameter between groups treated with esmolol vs. untreated animals were observed (LD) (Figure 2A).

The effects of esmolol on coronary artery composition (medial and adventitial layers) are shown in Figure 3 and Figure 4. Forty-eight hours of treatment with esmolol induced a significant reduction in wall thickness and cell count in the SHR-E 48 h group compared with the SHR 48 h group. These effects persisted after treatment withdrawal (SHR-E 7 d group vs. SHR 7 d group, and SHR-E 1 m group vs. SHR 1 m group). There were no significant differences in the medial layer cell density between esmolol-treated groups with respect to untreated animals. Esmolol induced a significant reduction in cell count in the adventitial layer, and cell density was significantly minor in the SHR-E 48 h group with respect to the SHR 48 h group; however, these changes did not persist after esmolol suppression. There were no significant differences in the adventitial layer wall thickness between esmolol-treated groups with respect to untreated animals.

### 3.4. Esmolol Improves Plasma Redox Status, and It Persists after Treatment Withdrawal

Esmolol induced a significant increase in GSH in the SHR-E 48 h group with respect to the SHR 48 h group, and this effect persisted after treatment withdrawal (SHR-E 7 d group vs. SHR 7 d group, and SHR-E 1 m group vs. SHR 1 m group) (Figure 5A). Esmolol induced a significant increase in plasma nitrates in the SHR-E 48 h group with respect to the SHR 48 h group. Treatment withdrawal increased this biomarker in the SHR-E 7 d group but decreased it in the SHR-E 1 m group; however, the values were still lower than in the respective SHR control groups (Figure 5B).

Both peroxidation lipid (MDA + HNE) and carbonyls were significantly reduced in the SHR-E 48 h group with respect to the SHR 48 h group. Treatment withdrawal increased both oxidative damage biomarkers in the SHR-E 7 d and SHR-E 1 m groups, but the values were still lower than those of the SHR 7 d and SHR 1 m groups, respectively, (Figure 5C,D).

## 4. Discussion

This is the first study to show regression of coronary artery remodeling after a short treatment with an antihypertensive drug (esmolol) that persisted following treatment withdrawal.

Hypertension leads to adverse cardiovascular remodeling and, therefore, increases the incidence of cardiovascular events (myocardial ischemia, arrhythmias, heart failure and sudden death) [22,23]. Regression of cardiovascular remodeling is an objective of antihypertensive therapy, reducing cardiovascular events. Esmolol is a cardioselective β_1_-receptor blocking agent with the beneficial effects of β-blockers but without the detrimental effects of long-acting agents due to its pharmacokinetics (ultra-short half-life of 9min, rapid hydrolysis by plasma esterases and full therapeutic effect in 5 min) [14]. The efficacy of esmolol has been demonstrated in clinical practice in the treatment of supraventricular arrhythmias and perioperative tachycardia and hypertension [24]. In the literature, we found that esmolol has cardioprotective effects in a number of settings (acute myocardial infarction, during cardioplegic cardiac arrest and after cardiopulmonary bypass) [15,25] but not in cardiovascular remodeling. Our group demonstrated vascular protection by esmolol in an animal model of arterial hypertension and coronary damage and its persistence after discontinuation of treatment.

Several studies investigated regression artery remodeling with antihypertensive drugs [5,6,7,8,9,26,27,28,29]. However, only a few showed vascular protection after withdrawal of treatment [30]. Nebivolol induced a reduction in aortic media thickness and improved the acetylcholine-induced relaxant response after chronic treatment (6 months) in SHRs. These effects were maintained 3 months after drug withdrawal [30].

A few studies showed regression of hypertensive vascular remodeling after the administration of drugs as a short-term treatment [10,11,28]. Nifedipine, after 1 week of administration, attenuated aortic wall thickness and the cross-sectional area, and improved endothelium-dependent relaxation in SHRs; however, all parameters returned to control SHR values after a very short withdrawal from the drug (72 h) [28]. In our study, esmolol produced the same effects as nifedipine on the coronary artery with short-term treatment (48 h); however, these effects persisted after a 1-month withdrawal.

Reduction in blood pressure has been related to the regression of vascular remodeling. Chronic treatment with nebivolol reduced systolic blood pressure in SHRs, which increased progressively after withdrawal without reaching the values of the SHR controls. These changes were associated with aortic remodeling [30]. In our study, short-term treatment with esmolol decreased systolic blood pressure; however, this parameter returned to basal levels after withdrawal. Therefore, vascular protection by esmolol is not restricted to short-term blood pressure modulation. A blood-pressure-independent effect is also suggested in previous studies about vascular remodeling [11].

Hypertension-induced vascular dysfunction may occur as a contributing factor or as a consequence of vascular remodeling caused by chronically elevated systemic blood pressure [31]. The effect of 48 h of treatment with esmolol on vascular remodeling is related to increased nitric oxide bioavailability and a reduction in ADMA concentration in plasma [10,11]. Nitric oxide shows vasorelaxing and antiproliferative properties [32]. Therefore, esmolol improved the response of endothelium-dependent vasodilation in the coronary artery, and it decreased the contractile function (inhibiting SMC proliferation) as a result of the increased bioavailability of nitric oxide.

In the present study, we explored plasma redox status following the administration and withdrawal of esmolol in SHRs because oxidative stress is also associated with vascular remodeling in hypertension [33]. In the present study, esmolol improved plasma redox status (decreased lipid peroxidation and carbonyls and improved/reduced glutathione and nitrates), and this improvement was maintained a month after the end of treatment. Some studies attributed antioxidant properties to esmolol but in other contexts (myocardial infarction) [34,35].

The present study has limitations. This study was designed to analyze maintenance over time of esmolol’s effect on the structure and function of coronary arteries in SHRs. Endothelial disease is highly variable in SHRs due to differences in artery type, sex, age and methodology used to study vascular function [36]. The results of our study apply only to the left anterior descending coronary artery in 14-month-old male SHRs. However, we need the translation of vascular protection by esmolol from animal experiments to clinical practice using clinical trials. Cardiovascular changes in SHRs are similar to those occurring in human hypertension; therefore, our observations with esmolol are predictive of effects in human patients [37].

## 5. Conclusions

In conclusion, our results show that the regression of coronary artery remodeling after a short treatment with an antihypertensive drug (esmolol) persists following withdrawal. The improvement in plasma oxidative status could be implicated in this effect.

## Figures and Tables

**Figure 1 antioxidants-11-02042-f001:**
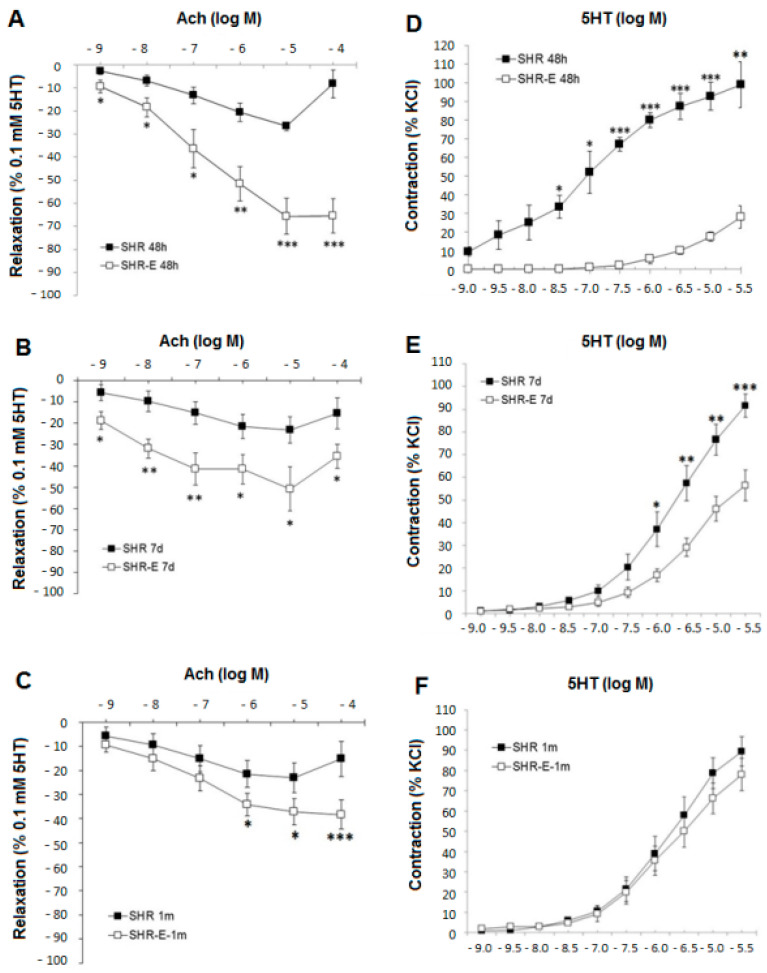
Concentration–response curves for acetylcholine on the coronary artery: (**A**) in treatment groups (48 h) and (**B**), (**C**) in withdrawal groups (7 days and 1 month). Concentration–response curves for serotonin on the coronary artery: (**D**) in treatment groups and (**E**), (**F**) in withdrawal groups (7 days and 1 month). SHRs, rats treated with vehicle; SHR-Es, rats treated with esmolol; *n* = 9 rats for each experimental group. Values are mean ± SEM. * *p* < 0.05, ** *p* < 0.01, *** *p* < 0.001 vs. SHRs.

**Figure 2 antioxidants-11-02042-f002:**
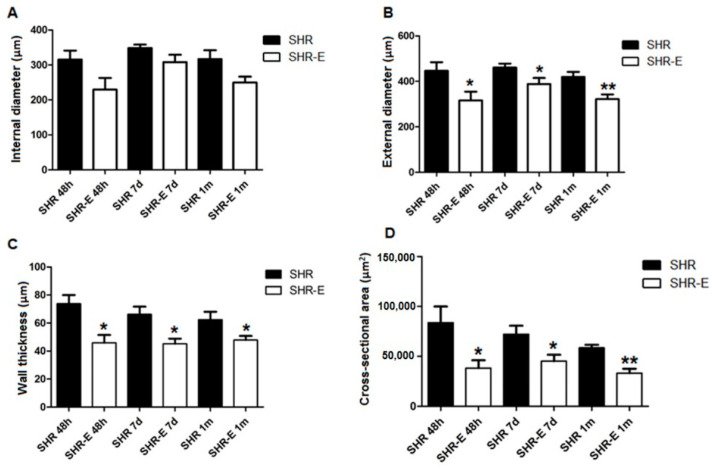
Structural parameters (medial + adventitial layers) of the coronary artery. Lumen diameter (**A**), external diameter (**B**), wall thickness (**C**), and cross-sectional (**D**) in treatment groups (48 h) and withdrawal groups (7 days and 1 month). SHRs, rats treated with vehicle; SHR-Es, rats treated with esmolol; *n* = 9 rats for each experimental group. Values are mean ± SEM. * *p* < 0.05, ** *p* < 0.01 vs. SHRs.

**Figure 3 antioxidants-11-02042-f003:**
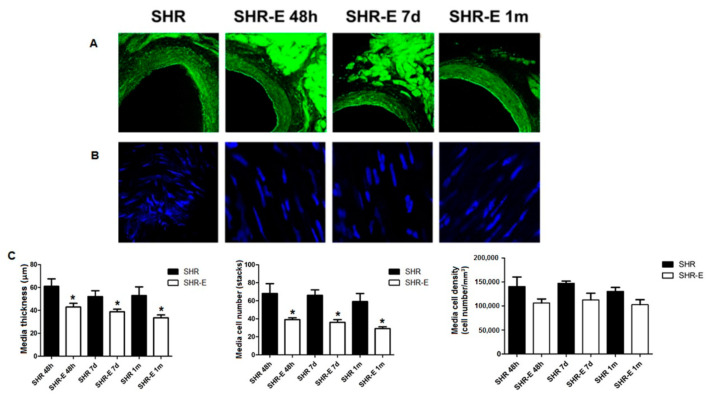
Medial layer of the left anterior descending coronary artery. Shown are rings (confocal microscopy images, ×20, zoom 2) (**A**), cell nuclei (confocal microscopy images, ×20, zoom 8) (**B**) and the composition of the medial layer (**C**) of the coronary artery in treatment groups (48 h) and withdrawal groups (7 days and 1 month). SHRs, rats treated with vehicle; SHR-Es, rats treated with esmolol; *n* = 9 rats for each experimental group. Values are mean ± SEM. * *p* < 0.05 vs. SHRs.

**Figure 4 antioxidants-11-02042-f004:**
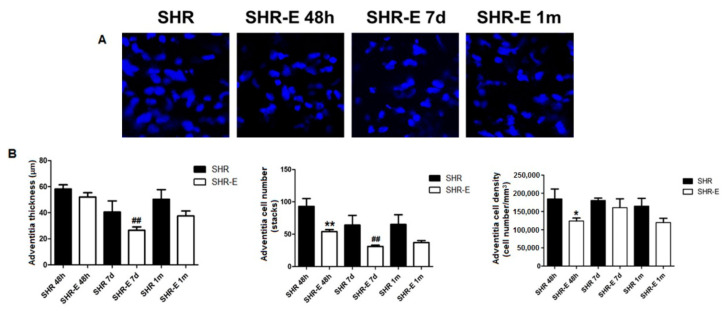
Adventitial layer of the left anterior descending coronary artery. Shown are cell nuclei (confocal microscopy images, ×20, zoom 8) (**A**) and the composition of the adventitial layer (**B**) of the coronary artery in treatment groups (48 h) and withdrawal groups (7 days and 1 month). SHRs, rats treated with vehicle; SHR-Es, rats treated with esmolol; *n* = 9 rats for each experimental group. Values are mean ± SEM. * *p* < 0.05, ** *p* < 0.01 vs. SHRs, ^##^
*p* < 0.01 vs. SHR-E 48 h group.

**Figure 5 antioxidants-11-02042-f005:**
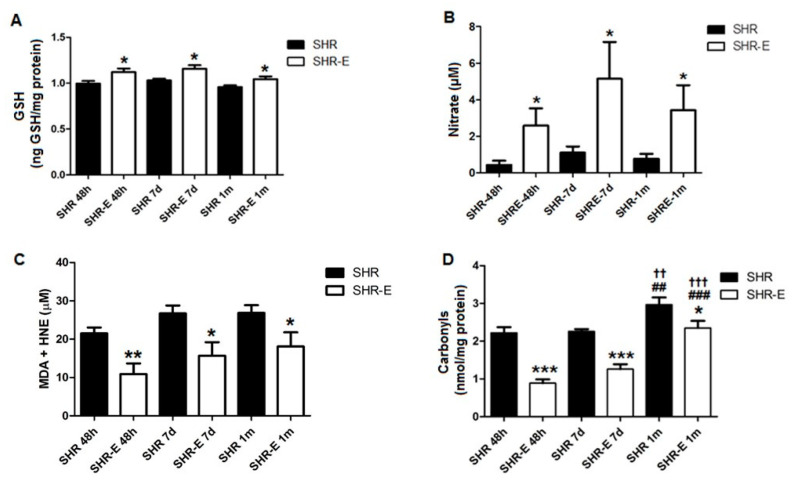
Oxidative status biomarkers. Reduced glutathione (GSH) (**A**), nitrate (**B**), malondialdehyde + 4-hydroxynonenal (MDA + HNE) (**C**) and carbonyls (**D**) in plasma of treatment groups (48 h) and withdrawal groups (7 days and 1 month). SHRs, rats treated with vehicle; SHR-Es, rats treated with esmolol; *n* = 8 rats for each experimental group. Values are mean ±SEM. * *p* < 0.05, ** *p* < 0.01, *** *p* < 0.001 vs. SHRs; ^##^
*p* < 0.01, ^###^
*p* < 0.001 SHR 1 m/SHR-E 1 m groups vs. SHR 48 h/SHR-E 48 h groups; ^††^
*p* < 0.01, ^†††^
*p* < 0.001 SHR 1 m/SHR-E 1 m groups vs. SHR 7 d/SHR-E 7 d groups.

**Table 1 antioxidants-11-02042-t001:** Blood pressure and heart rate.

	SHR (*n* = 9/Group)	SHR-E (*n* = 9/Group)
**Blood pressure (mmHg)**		
Treatment group	187 ± 20	139 ± 23 *
Withdrawal week group	183 ± 19	179 ± 16 #
Withdrawal month group	179 ± 15	189 ± 10 #
**Heart rate (beats/min)**		
Treatment group	380 ± 6	305 ± 10 *
Withdrawal week group	372 ± 9	369 ± 15 #
Withdrawal month group	389 ± 11	373 ± 7 #

Values are mean ± SEM. SHRs, rats treated with vehicle; SHR-Es, rats treated with esmolol. * *p* < 0.05 vs. SHRs; # *p* < 0.05 vs. SHR-Es after 48 h of treatment.

## Data Availability

Data is contained within the article.

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
