# Peer review of "Maintenance over Time of the Effect Produced by Esmolol on the Structure and Function of Coronary Arteries in Hypertensive Heart Diseases"

_antioxidants, 2022, doi:10.3390/antiox11102042_

Round 1
Reviewer 1 Report
Maintenance over time of the effect produced by esmolol on the structure and function of coronary arteries in hypertensive heart diseases by Martín-Oropesa et al.
In this manuscript, the authors continue on previously reported observations with respect to administration of esmolol. Esmolol was administered intravenously continuously for 48 hr in 14-month-old SHR (SHR-E). The authors observed esmolol treated animals to have, for example, deceased wall thickness. Esmolol treatments significantly improved the endothelium-dependent relaxation via ACh. Esmolol administration increased GSH and nitrate levels, while decreased reactive intermediates such as carbonyls and malondialdehyde.
Comments / feedback includes:
Line 23 A number at a beginning of a sentence should be written out. So, this may read better if … “Fourteen-month-old”
Line 33 ‘enhanced’ may read better if ‘enhancement’
Line 71 How many animals were housed per cage?
Line 74 Does the administration of diazepam and ketamine influence the experimental results? Did the authors consider CO2 to terminate the animals?
Line 75 Blood was drawn. How much? With an anti-coagulant? Route of blood removal?
Line 78 should it not read kg instead of k?
Line 78 From where was the esmolol procured and was this is saline when dosed? Please clarify experimental details
Line 90 need to lower case 4 in MgSO4 and 2 in H2O
Line 97 From where was the 5-HT procured and was it dosed in saline?
Line 130 Blood (how much, and was anti-coagulant used)
Line 134 From where was o-phthalaldehyde procured?
It would seem that an experimental procedure for this analysis is missing. Sufficient information needs to be provided in order to reproduce the work.
Line 139 volume may read better if it read ‘sample’
How long was the mixture incubated for?
Line 142 From where was the vanadium (III) chloride procured
Line 143 Significant figures ‘1 M’ should read ‘1.0 M’
Line 146 1 hour should read “1.0 hour”
Line 147 How was standard curve prepared?
Line 329 Seems like a literature reference is needed here
Overall the experimental findings are interesting and paper organized and well written; so, worthy of publication after minor issues are address (per above).
Author Response
Response to Reviewer 1 Comments
Please see attached file. Changes in the manuscript are in red
Line 23 A number at a beginning of a sentence should be written out. So, this may read better if … “Fourteen-month-old”
Response: We make the correction in the manuscript
Line 33 ‘enhanced’ may read better if ‘enhancement’
Response: We make the correction in the manuscript
Line 71 How many animals were housed per cage?
Response: 4 animals per cage were housed. We include this data in the text
Line 74 Does the administration of diazepam and ketamine influence the experimental results? Did the authors consider CO2 to terminate the animals?
Response: Ketamine is usually employed in cardiovascular research (Kawahara Yuji Y et al. J Pharmacol Sci 2005;99:95-104) to induce sedation and inmobility, and it is usually used in combination with diazepam to improve muscle relaxation and prolong analgesia (Yang XP et al. Am J Physiol Heart Circ Physiol 1999;277:H1967-H1974). All animals received the same anesthetic protocol, maintaining the hamodynamic stability. This is important so that anesthetics do not influence the experimental results.
In the present study, at the end of treatment, the rats were sedated with diazepam and ketamine intraperitoneal injection. Thereafter, rats were killed by decapitation. Decapitation is an accepted method of small animal euthanasia. Animal stress is avoided by anesthetizing prior to decapitation.
Line 75 Blood was drawn. How much? With an anti-coagulant? Route of blood removal?
Response: blood was obtained by puncture of the tail vein (1.2 mL), and it was transferred to tubes with heparin, 5000 units.
We make the correction in the manuscript.
Line 78 should it not read kg instead of k?
Response: We make the correction in the manuscript
Line 78 From where was the esmolol procured and was this is saline when dosed? Please clarify experimental details
Response: We use esmolol: Brevibloc 10 mg/ml solution for infusion (Baxter).
Brevibloc 10 mg/ml solution for infusion contains 10 mg esmolol hydrochloride per ml. Each 250 ml bag contains 2500 mg of esmolol hydrochloride. It is a solution with esmolol hydrochloride and saline for intravenous infusion.
Line 90 need to lower case 4 in MgSO4 and 2 in H2O
Response: We make the correction in the manuscript
Line 97 From where was the 5-HT procured and was it dosed in saline?
Response: Sigma-Aldrich. It was dosed with bidistilled water. We include this information in the manuscript
Line 130 Blood (how much, and was anti-coagulant used)
Response: Blood 1.2 mL, with heparin. We include this information in the manuscript
Line 134 From where was o-phthalaldehyde procured?
It would seem that an experimental procedure for this analysis is missing. Sufficient information needs to be provided in order to reproduce the work.
Response: Sigma-Aldrich. We include this information in the manuscript
Line 139 volume may read better if it read ‘sample’
How long was the mixture incubated for?
Response: In brief, a 100 µL plasma sample was incubated with 10 µL N-ethylmaleimide 150 mM, during 5 minutes at room temperature We include this information in the manuscript
Line 142 From where was the vanadium (III) chloride procured
Response: Sigma-Aldrich. We include this information in the manuscript
Line 143 Significant figures ‘1 M’ should read ‘1.0 M’
Response: We make the correction in the manuscript
Line 146 1 hour should read “1.0 hour”
Response: We make the correction in the manuscript
Line 147 How was standard curve prepared?
Response: The standard curve (0-100 µM) was prepared with a nitrate solution 100 µM and diluted in Methanol:H2O. We include this information in the manuscript
Line 329 Seems like a literature reference is needed here
Response: Frohlich ED. Is the spontaneously hypertensive rat a model for human hypertension? J Hypertension suppl 1986;4:S15-S19.
We include this information in the manuscript
We appreciate very much your suggestions and comments for improving the quality of the paper.
Thanks for your corrections. We look forward to hearing from you.
Sincerely,
Dr. Begoña Quintana-Villamandos MD, PhD
Reviewer 2 Report
This paper provides new insights into the effect of esmolol on coronary remodeling in a 14 month old hypertensive SHR rat model suggesting an interesting therapeutic potential for this known drug.
The overall picture is well structured and challenging despite the intrinsic complexity of the topic. Nonetheless I would suggest more emphasis on the correlation between the structural part of the paper and the functional one. In particular the key point of endothelial function should be made more prominent.
Specific comments on your work are the following:
Age of the rats: SHR rats spontaneously develop hypertension from 10 weeks of age. Why did you choose to set up your experimental procedure with quite old rats? Have you preliminary data to explain your choice?
Dose of the drug: The dose of esmolol used (300 μg/kg/min) seems high as the usual dose in humans is usually 50- 200 μg/kg/min. Have you preliminary data supporting your choice?
My other comments relate only to minor points:
1. Pag 6, lane 207: experiments on artery geometry should should be explained more clearly perhaps with the support of a scheme
2. Pag 6, lane 230: “Esmolol induced a significant reduction in cell count…” The sentence is not clear. It seems that reducing cell count, cell density increases.
3. Check minor errors as in pag 1. Lane 23: “is persistence” should be “is persistent”.
Author Response
Response to Reviewer 2 Comments
Please see attached file. Changes in the manuscript are in red
The overall picture is well structured and challenging despite the intrinsic complexity of the topic. Nonetheless I would suggest more emphasis on the correlation between the structural part of the paper and the functional one. In particular the key point of endothelial function should be made more prominent.
Response: Hypertension involve attenuated vascular relaxation and increased contraction in the artery. The effect of 48 h of treatment with esmolol on vascular remodeling is related to increased nitric oxide bioavailability. NO shows vasorelaxing and antiproliferative properties. Therefore, esmolol improved the response endothelium-dependent (vasodilatation) in the coronary artery owing to increased bioavailability of NO. NO is an important regulator of vascular remodeling, inhibiting SMC proliferation. Thus, reduction in SMC after treatment with esmolol might contribute to decreased contractile function. We include these comments in the manuscript (Discussion).
Specific comments on your work are the following:
Age of the rats: SHR rats spontaneously develop hypertension from 10 weeks of age. Why did you choose to set up your experimental procedure with quite old rats? Have you preliminary data to explain your choice?
Response: At 14 months of age, SHR present compensated left ventricular hypertrophy (Brooks WW et al. L-arginine fails to prevent ventricular remodeling and heart failure in the spontaneously hypertensive rat,” American Journal of Hypertension, vol. 22, no. 2, pp. 228–234, 2009], which is associated with functional and structural alterations of the coronary artery (Tanaka M et al. Quantitative analysis of myocardial fibrosis in normals, hypertensive hearts, and hypertrophic cardiomyopathy, British Heart Journal, vol. 55, no. 6, pp. 575–581, 1986).
On the other hand, aging seems to be an important factor in studying endothelial dysfunction (ED). ED was observed mainly in adult and aged (older than 25 weeks) and not in young SHR (Bernatova I et al. Endothelial dysfunction in spontaneously hypertensive rats: focus on methodological aspects,” Journal of Hypertension, vol. 27, pp. S27–S31, 2009).
Dose of the drug: The dose of esmolol used (300 μg/kg/min) seems high as the usual dose in humans is usually 50- 200 μg/kg/min. Have you preliminary data supporting your choice?
Response: In this study we have used infusion of esmolol of 300 µg/kg/min, because this dose have been used in clinical practice. And 300 µg/kg/min is the maximun dose recommended (ttp://www.aemps.gob.es/). This is administration rate related to the human studies:
- Wiest DB et al. Clinical pharmacokinetics and therapeutic efficacy of esmolol. Clin Pharmacokinet. 2012 Jun 1;51(6):347-56
- Yu SK, Tait G, Karkouti K, Wijeysundera D, McCluskey S, Beattie WS. The safety of perioperative esmolol: a systematic review and meta-analysis of randomized controlled trials. Anesth Analg 2011; 112:267-281.
- Garnock-Jones KP. Esmolol: a review of its use in the short-term treatment of tachyarrhythmias and the short-term control of tachycardia and hypertension. Drugs 2012; 72:109-132.
- Kirshenbaum JM, Kloner RF, McGowan N, Antman EM. Use of an ultrashort-acting beta-receptor blocker (esmolol) in patients with acute myocardial ischemia and relative contraindications to beta-blockade therapy. J Am Coll Cardiol, 1988; 12:773-780.
My other comments relate only to minor points:
1.Pag 6, lane 207: experiments on artery geometry should be explained more clearly perhaps with the support of a scheme
Response: We explain again the experiments on artery geometry in the manuscript.
2.Pag 6, lane 230: “Esmolol induced a significant reduction in cell count…” The sentence is not clear. It seems that reducing cell count, cell density increases.
Response: Esmolol induced a significant reduction in cell count in the adventitial layer, and cell density was significantly minor in the SHR-E 48hr group with respect to the SHR 48hr group.
We include these comments in the manuscript.
3.Check minor errors as in pag 1. Lane 23: “is persistence” should be “is persistent”.
Response: We make the correction in the manuscript
We appreciate very much your suggestions and comments for improving the quality of the paper.
Thanks for your corrections. We look forward to hearing from you.
Sincerely,
Dr. Begoña Quintana-Villamandos MD, PhD